# Modeling of Self-Aligned Selector Based on Ultra-Thin Metal Oxide for Resistive Random-Access Memory (RRAM) Crossbar Arrays

**DOI:** 10.3390/nano14080668

**Published:** 2024-04-12

**Authors:** Mikhail Fedotov, Viktor Korotitsky, Sergei Koveshnikov

**Affiliations:** Institute of Microelectronics Technology and High-Purity Materials, Russian Academy of Science (IMT RAS), 6 Academician Ossipyan Str., Moscow District, Chernogolovka 142432, Russia

**Keywords:** analog neural networks, crossbar, dielectric layers, memristor circuits, resistive RAM, selectors

## Abstract

Resistive random-access memory (RRAM) is a crucial element for next-generation large-scale memory arrays, analogue neuromorphic computing and energy-efficient System-on-Chip applications. For these applications, RRAM elements are arranged into Crossbar arrays, where rectifying selector devices are required for correct read operation of the memory cells. One of the key advantages of RRAM is its high scalability due to the filamentary mechanism of resistive switching, as the cell conductivity is not dependent on the cell area. Thus, a selector device becomes a limiting factor in Crossbar arrays in terms of scalability, as its area exceeds the minimal possible area of an RRAM cell. We propose a tunnel diode selector, which is self-aligned with an RRAM cell and, thus, occupies the same area. In this study, we address the theoretical and modeling aspects of creating a self-aligned selector with optimal parameters to avoid any deterioration of RRAM cell performance. We investigate the possibilities of using a tunnel diode based on single- and double-layer dielectrics and determine their optimal physical properties to be used in an HfOx-based RRAM Crossbar array.

## 1. Introduction

Resistive random-access memory (RRAM) is a promising candidate for advanced non-volatile memory due to its scalability, high write and erase speeds, low power consumption and simple design [1,2,3]. Owing to its outstanding properties, RRAM is not only considered as the standout candidate for emerging memory technologies [4] but also offers significant advantages for neuromorphic computing [5,6].

An RRAM cell is a simple two-terminal memory device. The memory effect is based on the reversible transition from a high-resistance state (HRS) to a low-resistance state (LRS) and back when the polarity of the applied voltage is changed. These resistive state transitions can be achieved by formation/growth and disruption of a conductive filament (CF) through an insulating layer, in particular through a transition metal oxide [7].

Metal-oxide-based RRAM demonstrates excellent scaling potential down to <10 nm [8]. The scalability of an RRAM cell down to a size of a few nanometers [9] is possible due to the filamentary nature [7] of resistive switching for a variety of metal oxides, including hafnium oxide [7,10], titanium oxide [11,12], tantalum oxide [13,14], zirconium dioxide [15,16] and other metal-based oxides. A nano-size conductive filament, which controls switching from HRS to LRS, makes the cell conductivity insensitive to cell size [17], thus providing potential strong memory cell scalability.

Crossbar RRAM arrays are crucial for creating the next generation of large memory arrays, Crossbar arrays for matrix multiplication and analogue neuromorphic networks [18,19]. However, in Crossbar structures, an additional selector device is required to eliminate sneak path currents and provide accurate read and write operations [20,21,22,23,24]. There are a variety of selector devices, including transistors [25], tunnel diodes [21,26,27], varistors [28], mixed-conductivity devices (MIEC) [29], punchthrough diodes [30], electromechanical diodes [31] and Zener diodes [32]. However, all these options share one common issue: their area is larger than that of a memory cell. Unlike RRAM, a selector’s conductivity depends on its area, thus making the selector a limiting factor in large Crossbar RRAM array scaling. A promising class of selectors allowing for a progressive increase in memory array capacity is the metal–insulator–metal (MIM) structure, which is self-aligned with a memory cell [21,27,33]. This particular type of selector is a tunnel diode with a single-layer or a double-layer dielectric. The area of the selector device is equal to the area of the memory cell in a 1S1R structure (Figure 1). This design has a clear advantage: among all possible options, it is a self-aligned selector that has the smallest area. Therefore, a self-aligned selector allows one to achieve the highest storage density in a memory array. The other options such as MOSFETs require additional power and space on the chip. The vertical Gate-All-Around nano-pillar transistor that was proposed in [34] allows one to achieve a 4F^2^ footprint (where F is a minimum cell feature size); however, this compact 1T-1R architecture is significantly more complex for fabrication, as compared to a simple MIM architecture.

The development of self-aligned selectors is underway in several research groups [21,27,33], with an emphasis on empirical research. Our approach is based on theoretical calculations of the current–voltage characteristics of numerous insulating films as a function of their thickness, composition, band gap width and dielectric constant. We theoretically determine an optimal bilayer structure and the geometry of a selector device. This study presents theoretical calculations of conductivity in selector MIM structures with single-layer and double-layer dielectrics. Based on the electrical properties of the hafnium oxide-based RRAM cell, we determine optimal configurations of a self-aligned selector for a Crossbar structure with high storage density.

## 2. Materials and Methods

A self-aligned selector must have a number of important properties:

(1)The current–voltage characteristics are diode-like [22], without hysteresis during forward and reverse voltage sweep; i.e., the selector material should be free of shallow and deep traps, which are able to capture and hold negative charge.(2)Selector resistance at a programming voltage (typically of 0.5 V to 1 V) should be in a range of 10^2^ to 10^4^ Ω, depending on the memory cell resistance in its low-resistance state (LRS) [23].(3)Selector resistance at a reading voltage (typically 0.1 V) should be in a range of 10^4^ to 10^6^ Ω, depending on the memory cell resistance in its high-resistance state (HRS) [24].

These requirements are based on the resistive switching characteristics of hafnium-oxide-based RRAM published in [7,10], as well as on our own experimentally measured resistive switching characteristics of the 1 transistor-1 resistor (1T1R) structures with a 10 nm thick HfOx film and TaN metal electrodes [35]. The HRS and LRS currents are extracted from the current–voltage characteristics of the RRAM elements (Figure 2a), measured at various values of the compliance current (I_compl_) and controlled by the integrated field-effect transistor during the transition from HRS to LRS (Set). Both LRS and HRS currents increase with an increasing I_compl_ value. The LRS resistance as a function of the compliance current is presented in Figure 2b. The data for hafnium-oxide-based RRAM reported in [7,10] are also presented in Figure 2b for comparison. Thus, the data in Figure 2b define the current operation field for the selector to be used for hafnium-oxide-based RRAM. It should be noted that operation fields for other transition metal oxides are quite similar to that of hafnium oxide due to the filamentary nature of their resistive switching.

In addition to the compliance current that affects the HRS current, the maximum voltage used during the transition from LRS to HRS (Reset) can also control the HRS current. The switching characteristics obtained on the same device but with various values of the maximum Reset voltage are presented in Figure 3a. The dependence of the HRS current measured at a reading voltage of 0.1 V on the maximum Reset voltage is shown in Figure 3b.

During a programming step, the selector must not interfere with the transition of the memory cell to a low-resistance state. This implies that resistance is to be lower than the resistance of a memory cell at Set voltage. At the same time, during the reading operation, spontaneous switching of the cell from the high-resistance state (HRS) should not occur. The selector, in this case, should work as a reverse-biased diode, the resistance of which should be much greater than the resistance of the memory cell in the high-resistance state.

A physical model describing the conductivity in nanoscale dielectric structures is required to create an optimal self-aligned selector. The conductivity mechanism of selector based on MIM structure depends on the type of dielectric, its thickness and bandgap width. It can utilize Poole–Frenkel conductivity, Fowler–Nordheim tunneling [36], direct tunneling and trap-assisted tunneling [37]. It was demonstrated in [36] that several mechanisms can work simultaneously, e.g., the Poole–Frenkel and Fowler–Nordheim conductivity. The conductivity model can also vary depending on the magnitude of the voltage applied to the dielectric [36]. However, in most single-layer dielectrics with nanometer-scale thicknesses and voltages in the order of 1 V (typical writing voltage in RRAM), the main mechanism of conductivity derived by the Wentzel–Kramers–Brillouin (WKB) approximation [38] is a form of direct tunneling through a rectangular potential barrier (Figure 4, Equations (1) and (2)), which, at high voltages, smoothly transforms into Fowler–Nordheim tunneling (Equation (3)) [38]. This model is confirmed by both theoretical [39] and experimental [21,40] studies. Some additional possible conduction mechanisms, such as trap-assisted tunneling and Poole–Frenkel conduction, are out of the scope of this work, as they could be related to specific dielectric materials with electron traps, such as Al_2_O_3_ [36].

The potential barrier that exists between an electrode and a dielectric layer has two important parameters: the width, which equals the thickness of a corresponding dielectric layer, and height, that, in terms of energy, is equal to the difference between electron affinity of dielectric and work function of the electrode. In this study, we consider only the structures with metal electrodes made of the same material.

In case of a single-layer dielectric and for low voltages (V < φ), the WKB approximation [38] yields classical equation of direct tunneling:(1)$$J=J_{0}\left(\left(\varphi -\frac{eV}{2}\right)\times \exp\left(-A\sqrt{\varphi -\frac{eV}{2}}\right)-\left(\varphi +\frac{eV}{2}\right)\times \exp\left(-A\sqrt{\varphi +\frac{eV}{2}}\right)\right)$$
where J is current density, φ is potential barrier height, e is electron charge, V is applied voltage and J_0_ and A are the following constants [38]:(2)$$J_{0}=\frac{e}{2\pi hd^{2}}\;;A=\frac{4\pi d\sqrt{2m}}{h}$$
where e is electron charge, h is Planck’s constant, d is dielectric thickness and m is electron’s effective mass.

For higher voltages (V > φ), the potential barrier shape is changed from rectangular to triangular [38], and the effective width of the barrier is reduced. In this case, the WKB approximation yields
(3)$$J=\frac{2.2e^{3}F^{2}}{8\pi h\varphi }\left(\exp\left(-\frac{8\pi }{2.96heF}\sqrt{2m\varphi^{3}\;}\right)-\left(1+\frac{2\mathrm{eV}}{\varphi }\right)\exp\left(-\frac{8\pi }{2.96heF}\sqrt{2m\varphi^{3}\;\left(1+\frac{2\mathrm{eV}}{\varphi }\right)}\right)\right)$$
where electric F is field in dielectric. This equation describes the Fowler–Nordheim tunneling.

We used the WKB approximation to calculate conductivity of the step-shaped barriers in double-layer dielectric structures. Based on Simmons’ research [38], we calculated the direct tunneling equation for a bilayer structure with a step-shaped potential barrier (Figure 5).

In case of potential barrier that consists of several dielectric layers, the total tunneling probability is equal to product of tunneling probabilities through each layer. According to WKB approximation, the tunneling probability through i-th layer is identical to the case of single arbitrary-shaped barrier [38]:(4)$$P_{i}=\exp\left[-2\sqrt{\frac{2m_{i}}{\hbar^{2}}}\overset{x_{i}}{\underset{x_{i-1}}{\int }}\sqrt{U_{i}\left(x\right)-E}dx\right]$$
where x_i−1_ and x_i_ are the boundaries of i-th layer, U_i_ is the shape of i-th barrier as a function of x, m_i_ is electron’s effective mass in i-th layer and E is kinetic energy of incident electron. In our case of rectangular potential barrier,
(5)$$U_{i}\left(x\right)=\varphi_{i}-\frac{eV_{i}x}{d_{i}}$$
where V_i_ is voltage drop in i-th layer, d_i_ is respective layer’s thickness and φ_i_ is the bottom energy of conduction band of i-th dielectric (Figure 6).

V_i_/d_i_ in Equation (5) is equal to electric field D_i_ in i-th layer and can be calculated as
(6)Di=Vidi=Vεi*∑j=1idjεj
where ε_i_ is dielectric constant of i-th layer. Substituting (5) and (6) into (4) yields
(7)Pi=exp[−4εi2mi*ℏ2×∑j=1i(djεj)3eV{(Ui−eVi−E)3/2−(Ui−eVi+1−E)3/2}]
where m_i_ is electron effective mass in the i-th dielectric layer, V_i_ is voltage drop in the i-th layer, V is applied voltage, d_j_ is thickness of each dielectric, ε_j_ is dielectric constant of a respective layer and E is kinetic energy of incident electron. Equation (7) is valid under the assumption that all dielectric layers are homogenous—the dielectric constant ε_j_ of j-th layer is the same across the bulk of the layer. The tunneling probabilities P_i_ depend on the voltage drop across each layer, the material’s dielectric constant and thickness of the layer:(8)$$P\left(E\right)=\underset{i}{\prod }P_{i},$$

The total tunnel current density through the cell is determined by the difference (Equation (9)) between the forward flux (Equation (10)) caused by diffusion and the backward voltage-driven drift of electrons (Equation (11)).
(9)$$J_{total}=J_{forward}-J_{backward},$$
(10)$$J_{forward}=-\frac{me}{2\pi^{2}\hbar^{3}}\overset{0}{\underset{Ef}{\int }}P\left(E\right)EdE$$
(11)$$J_{backward}=-\frac{me}{2\pi^{2}\hbar^{3}}\overset{0}{\underset{Ef}{\int }}P\left(E\right)\left(E+eV\right)dE$$

This method can be used for all types of multilayer tunnel structures, regardless of number of dielectric layers and with arbitrary dielectric parameters, assuming that all dielectrics are homogenous. However, Equation (7) gives a correct result only for voltages lower than the height of potential barrier (U_i_ > eV_i_). For higher voltages, the conductivity is dominated by the highest barrier in multilayer structure, and in this case, one should use Equation (3). In case of a single-layer rectangular barrier, the integration of (9) yields the well-known Simmons Formula (1) [38]. We neglected the effect of the image forces occurring in potential barrier for the sake of simplifying the calculations.

## 3. Results

### 3.1. Single-Layer Selector

We simulated the I-V characteristics for dozens of single-layer dielectric structures with various electrode and dielectric materials. The best structures that meet the abovementioned requirements are presented in Figure 7. To verify the correctness of the employed model, we calculated the conductivity of the MIM selector based on the 2 nm thick Ta_2_O_5_ used in [21]. Figure 8 presents the experimentally measured [21] and calculated I-V characteristics of the ultra-thin tantalum-oxide-based selector. The comparison reveals good agreement for the measured and calculated currents in the −2 V to 2 V voltage range that covers the operating range for the selector.

According to the above-mentioned requirements for the selector, the steeper I-V curve results in better performance of the MIM selector. In Figure 8, the red dashed line represents the I-V characteristic of an optimal MIM selector with a minimal possible non-linearity and the current density in its open state. The stronger the current dependence on voltage and the higher the current density, the better the performance of the selector, and its size can be scaled down. We found that the desired characteristics of a selector are determined by the optimal values of height and width of a potential barrier between the electrode and dielectric (Figure 9a,b). Figure 9a shows the dependence of I-V curve non-linearity on the barrier’s height, while the thickness of the dielectric is fixed at 2 nm. The non-linearity of the I-V curve drops exponentially with an increase in barrier height. The thickness of the potential barrier affects the non-linearity in an opposite way. Figure 9b shows the dependence of I-V curve non-linearity on the barrier’s width (barrier height is fixed at 0.9 eV); an increase in dielectric layer thickness results in better non-linearity, which increases exponentially with barrier thickness. Therefore, a lower and thicker barrier results in a steeper I-V curve for selectors with a single-layer dielectric. In this case, the layer thickness cannot exceed 3 nm; otherwise, the current flowing through the selector in series with an RRAM cell is insufficient for the correct operation of the memory cell. The optimal barrier height for a 2 to 3 nm thick dielectric is found to be 0.3 to 0.5 eV. The materials presented in Figure 6 are the most satisfactory dielectrics to be used for self-aligned selectors.

### 3.2. Double-Layer Selector

Adding a second dielectric layer makes further I-V curve modulation possible, i.e., increasing its non-linearity without a decrease in conductivity. The barrier becomes step-shaped, with a narrow-band dielectric acting as a lower step and a wide-band dielectric acting as a higher step. By varying the height and width of each step, one can control both the I-V curve steepness and current density independently. We studied how both parameters depend on the barrier geometry (Figure 10, Figure 11, Figure 12 and Figure 13).

The dependence of the I-V curve on the barrier geometry presented in Figure 10, Figure 11, Figure 12 and Figure 13 reveals clear trends for the I-V curve non-linearity and the current density at 1 V (a typical write voltage for RRAM cell). The non-linearity of the I-V characteristics depends primarily on the narrow-band dielectric width and the height of its potential barrier. The non-linearity increases rapidly with an increasing barrier thickness for both positive and negative biases (Figure 14a), while the non-linearity dependence on the barrier height is sensitive to the voltage polarity (Figure 14b). The non-linearity dependence on the width and height of a wide-band dielectric is not so strong, as compared to the effect of a narrow-band dielectric (Figure 15a,b).

The current density depends on the thickness and barrier height of both dielectric layers. The current density increases with decreasing layer thickness (Figure 16a) and reducing barrier height (Figure 16b). The key factors controlling the current density are wide-band dielectric thickness (Figure 16a) and narrow-band dielectric barrier height (Figure 16b). However, an increasing dielectric thickness and barrier height result in decreasing current density. This, in turn, negatively affects selector scalability, as the selector with a lower conductivity requires a larger area to be effective for the write operation.

## 4. Discussion

As shown above, the main mechanism of conductivity for thin dielectrics in a low electric field is direct tunneling through a rectangular potential barrier. In this case, the current flowing through the dielectric layer can be described by the WKB approximation [38]. At a higher voltage, when the shape of a potential barrier becomes triangle-like, the conductivity is followed by the Fowler–Nordheim tunneling mechanism. The feasibility of this approach is verified by calculating the I-V characteristics for the 2 nm thick Ta_2_O_5_ layer and comparison with the experimental results obtained in [21]. The calculations performed for a variety of sub-3 nm thick single-layer dielectrics demonstrated that some of them can be used for self-aligned selectors (Figure 7).

We developed a compact analytical model of conductivity in multilayer tunnel barriers and utilized it to study the switching operation of a self-aligned selector with a double-layer tunnel barrier. The observed I-V curve dependencies on the thickness and barrier heights of dielectrics for double-layer barriers are critical for the determination of the optimal structure of an MIIM-diode selector. By increasing the number of dielectric layers and their thickness, the selector non-linearity is enhanced at the cost of current density. Therefore, an optimal balance between non-linearity and conductivity is of the greatest importance. We also emphasize that the parameters of a self-aligned selector to be employed in RRAM Crossbars strongly depend on the parameters of the RRAM cell connected in series to a selector. The most critical parameter of an RRAM cell is its resistances in the on and off states, which can be controlled by either the maximum current during forming or Set operation (Figure 2) and/or the maximum voltage during Reset (Figure 3). Furthermore, RRAM resistance depends on the materials used for RRAM cells and the thickness of the corresponding switching layer. The parameters of an optimal selector we propose in this work are developed on the basis of our previous study [35] on HfOx-based bipolar RRAM cells with a 10 nm HfOx layer and TaN electrodes. The selector allowing a RRAM cell to operate during bipolar resistive switching consists of 1.5 nm layers of Ta_2_O_5_ and Ga_2_O_3_ sandwiched between Ti electrodes (Figure 17). The barrier heights in this structure are 0.05 eV and 0.6 eV, respectively. Because of its asymmetric structure and optimized thickness and barrier height for each dielectric layer, it has 10-times higher non-linearity for the I-V curve compared to the best single-layer selectors, as well as 10-times higher current density in the open state (Figure 17).

The advantages of a selector with double-layer dielectric allows for scaling the cell in a Crossbar HfOx-based RRAM array down to 10^−^^3^ μm^2^ (Figure 18). In Figure 18, the red line represents the resistance of the above-mentioned selector with double-layer dielectric in the high-resistance state at a voltage equal to 0.1 V, and the black line is the resistance of the same selector in the low-resistance state at a voltage equal to 1 V. The operating range of the selector with an area of 10^−^^3^ μm^2^ covers the entire operation range of RRAM cells paired to the selector. Thus, it becomes theoretically possible to create a gigabit-scale memory array in an area of 1 cm^2^, which is impossible when using a selector with a single-layer dielectric.

## 5. Summary and Conclusions

The strong scalability of an RRAM cell due to the filamentary mechanism of resistive switching is the key advantage for the development of a high-density Crossbar memory array. A selector device that is self-aligned with a memory cell makes Crossbar array scalability possible. In this work, we addressed the theoretical and modeling aspects of creating a self-aligned tunnel diode-based selector with optimal parameters to avoid impacts on the resistive switching of an HfOx-based RRAM cell. The applicability of both single- and double-layer dielectrics for self-aligned selectors was demonstrated along with the determination of their optimal physical properties. Aside from RRAM crossbars, the MIIM diode optimization method we propose is equally applicable for optical rectification purposes in renewable energy applications, where both high non-linearity and high current density are required [40], as well as for other THz devices and prospective tunnel diodes.

However, the primary field where our findings are useful is the self-aligned selector technology for RRAM Crossbars, because for each type of RRAM cell, an individual selector is designed based on the cell’s electrical properties. The non-volatile resistance switching phenomenon has been reported for a variety of oxides and metal electrode materials [4]. High-density arrays made on both types of RRAM devices with conductive filaments, i.e., oxygen vacancies’ filament-based RRAM [7,10,11,12,16] and metal ion-based RRAM (or conductive bridge random-access memory) [14,41,42], require a proper selector to eliminate the sneaking current issue while not interfering with the RRAM cell performance.

Thus, our work can be used as a template for other researchers seeking to create RRAM memory arrays with low power consumption and high storage density. These devices are crucial for next-generation computers and the creation of analog neuromorphic networks.

## Figures and Tables

**Figure 1 nanomaterials-14-00668-f001:**
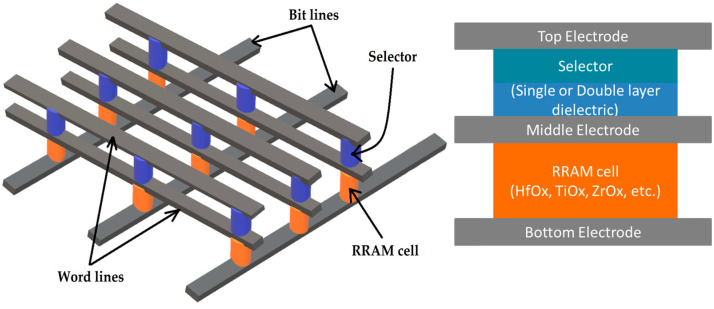
Three-electrode RRAM Crossbar with self-aligned selectors. Area of selector is equal to area of RRAM cell.

**Figure 2 nanomaterials-14-00668-f002:**
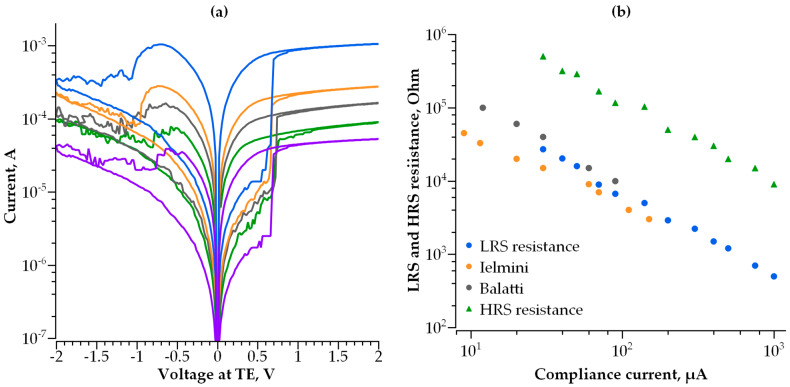
(**a**) Switching characteristics of HfO_x_-based RRAM obtained on the same device at various values of compliance current; (**b**) LRS and HRS resistance measured at a read voltage of 0.1 V as a function of compliance current, compared to [7,10].

**Figure 3 nanomaterials-14-00668-f003:**
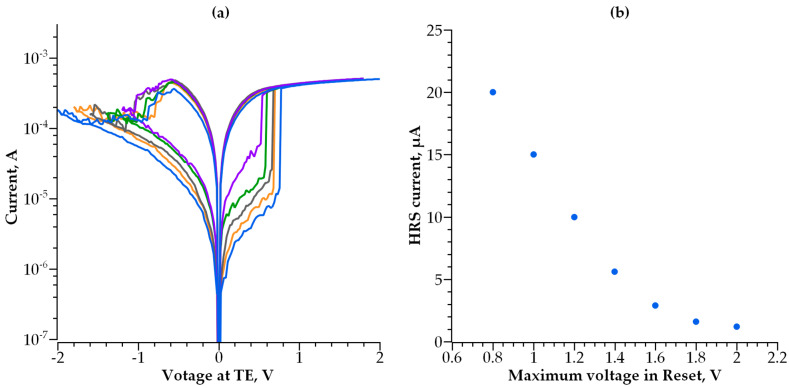
(**a**) Switching characteristics of HfOx-based RRAM obtained on the same device at various values of maximum voltage in Reset; (**b**) HRS current measured at a read voltage of 0.1 V as a function of maximum voltage in Reset.

**Figure 4 nanomaterials-14-00668-f004:**
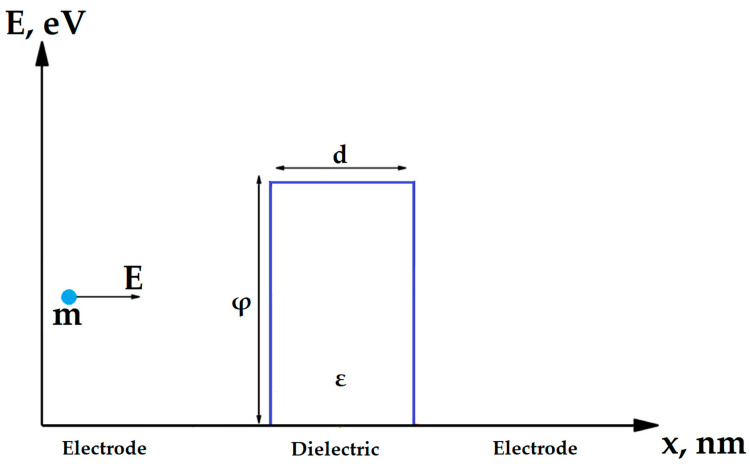
Band diagram of a tunnel diode selector with a single-layer dielectric.

**Figure 5 nanomaterials-14-00668-f005:**
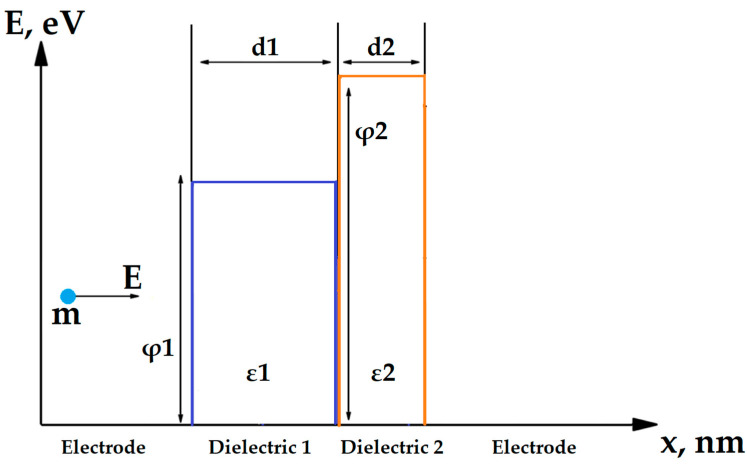
Band diagram of a tunnel diode selector with a double-layer dielectric.

**Figure 6 nanomaterials-14-00668-f006:**
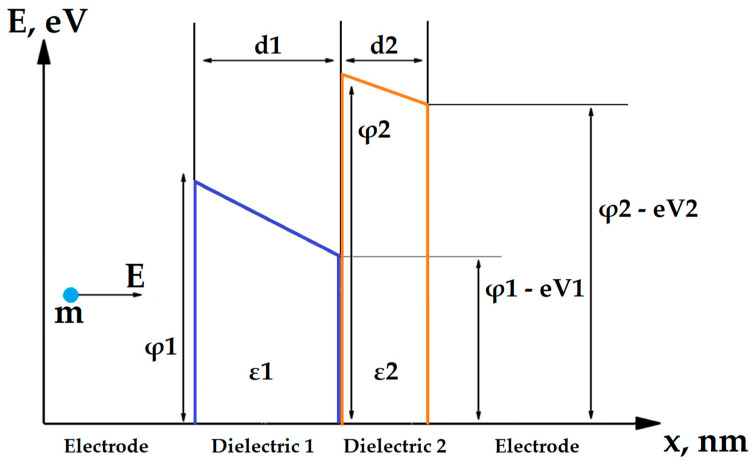
Band diagram of multilayer tunnel barrier under applied voltage.

**Figure 7 nanomaterials-14-00668-f007:**
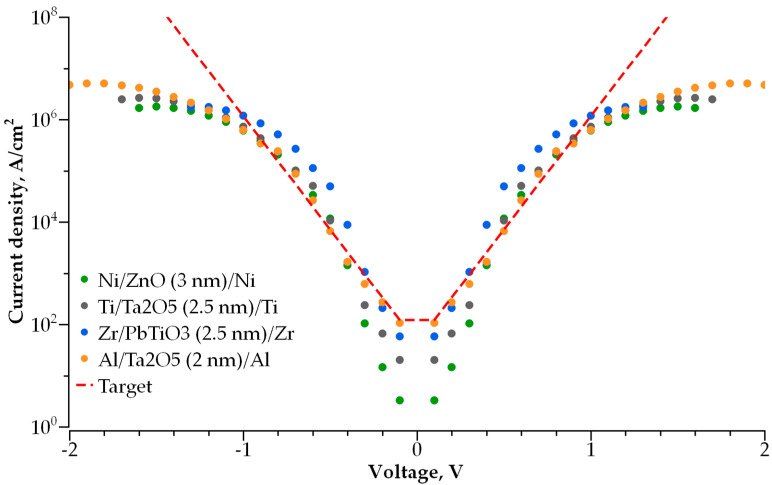
Simulated I-V curves of best single-layer (MIM) selector devices. Red dashed line is desired I-V curve.

**Figure 8 nanomaterials-14-00668-f008:**
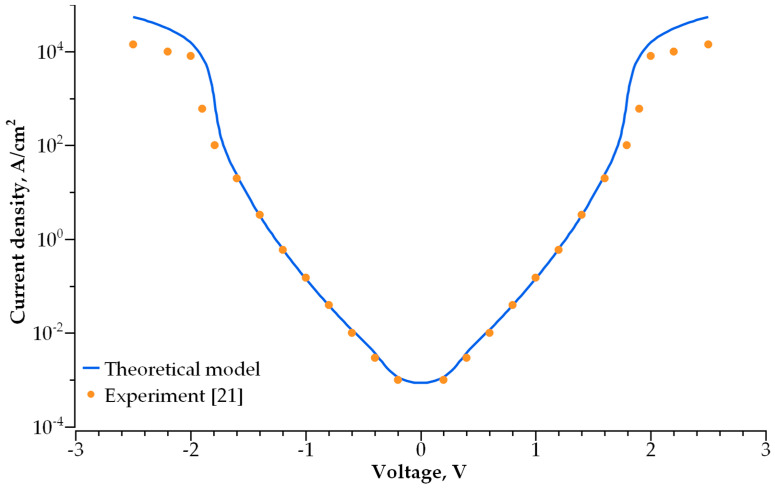
Comparison of the experimental [21] and modelled I-V characteristics for 2 nm thick Ta_2_O_5_ with a barrier height of 0.9 eV.

**Figure 9 nanomaterials-14-00668-f009:**
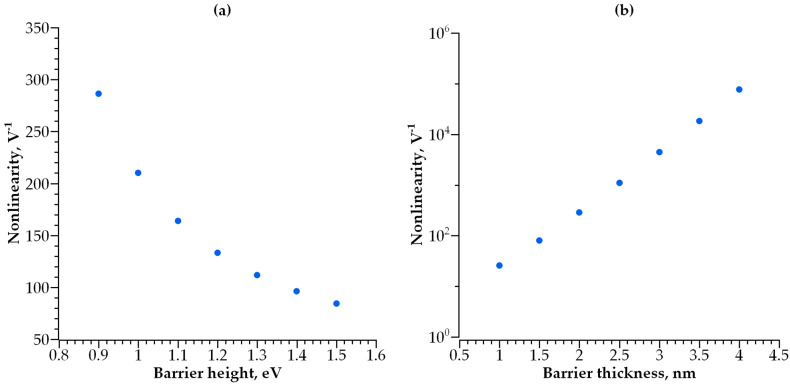
(**a**) I-V curve non-linearity dependence on barrier height, d = 2 nm; (**b**) I-V curve non-linearity dependence on barrier thickness, φ = 0.9 eV.

**Figure 10 nanomaterials-14-00668-f010:**
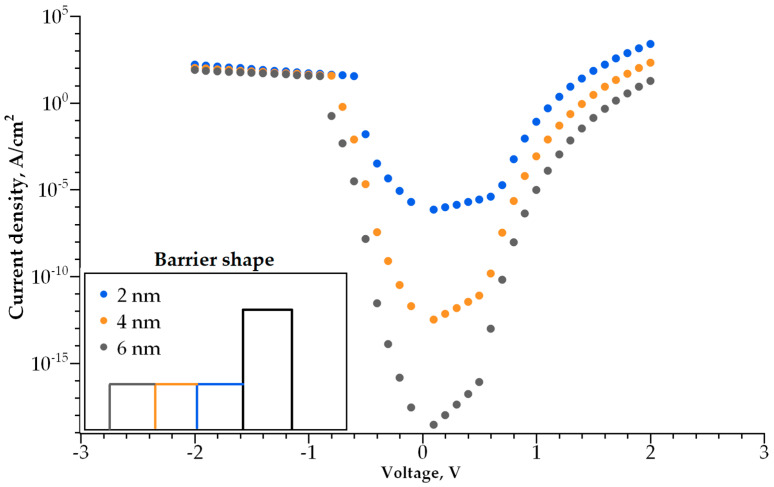
I-V curves for various thicknesses of narrowband dielectric layer. Inset shows the barrier shape.

**Figure 11 nanomaterials-14-00668-f011:**
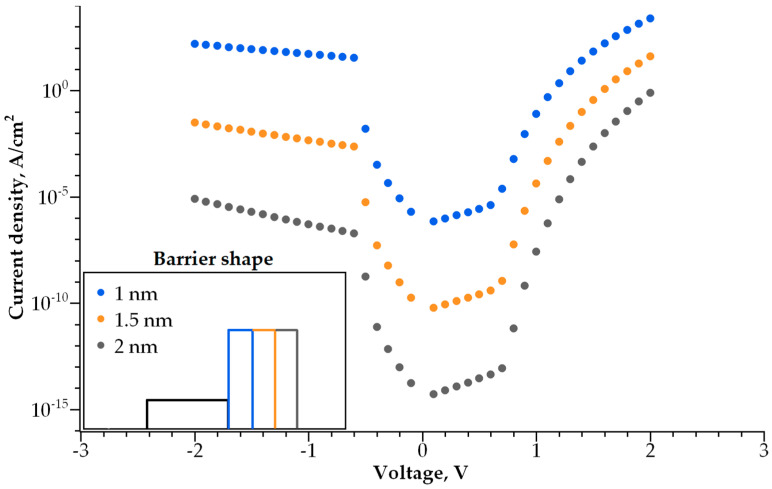
I-V curves for various thicknesses of wideband dielectric layer. Inset shows the barrier shape.

**Figure 12 nanomaterials-14-00668-f012:**
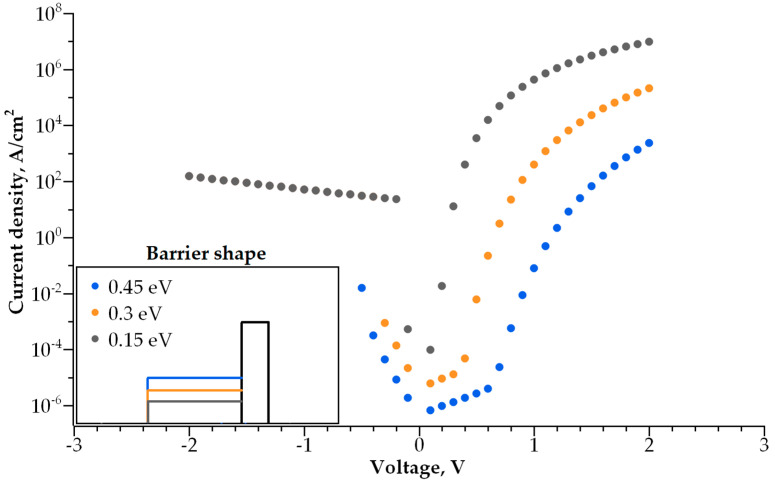
I-V curves for various barrier heights of the narrowband dielectric layer. Inset shows the barrier shape.

**Figure 13 nanomaterials-14-00668-f013:**
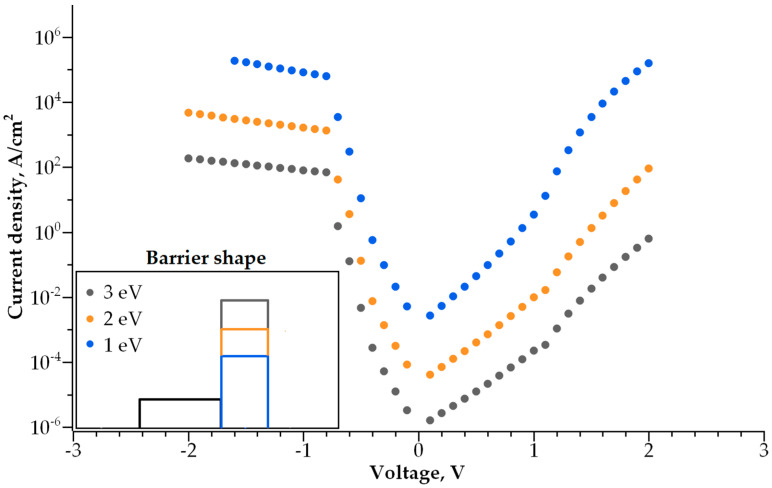
I-V curves for various barrier heights of the wideband dielectric layer. Inset shows the barrier shape.

**Figure 14 nanomaterials-14-00668-f014:**
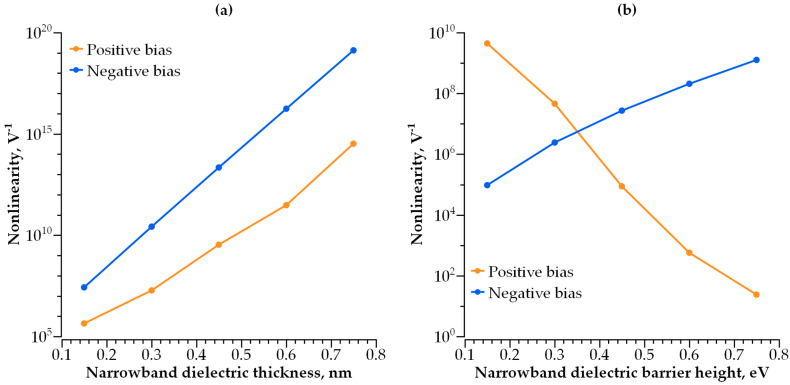
(**a**) I-V curve non-linearity dependence on narrow-band dielectric thickness; (**b**) I-V curve non-linearity dependence on barrier height of narrow-band dielectric.

**Figure 15 nanomaterials-14-00668-f015:**
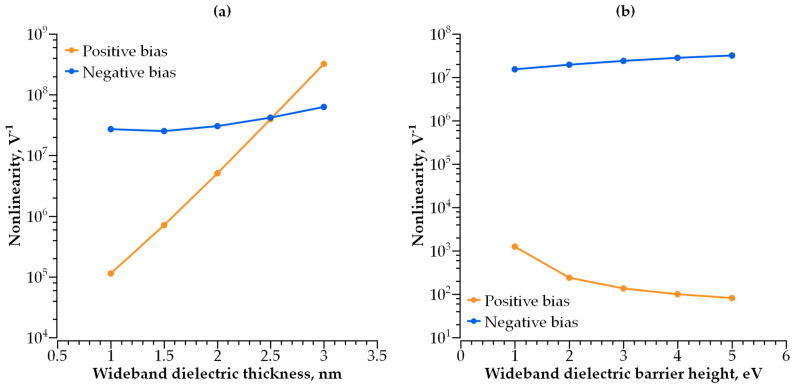
(**a**) I-V curve non-linearity dependence on wide-band dielectric thickness; (**b**) I-V curve non-linearity dependence on barrier height of wide-band dielectric.

**Figure 16 nanomaterials-14-00668-f016:**
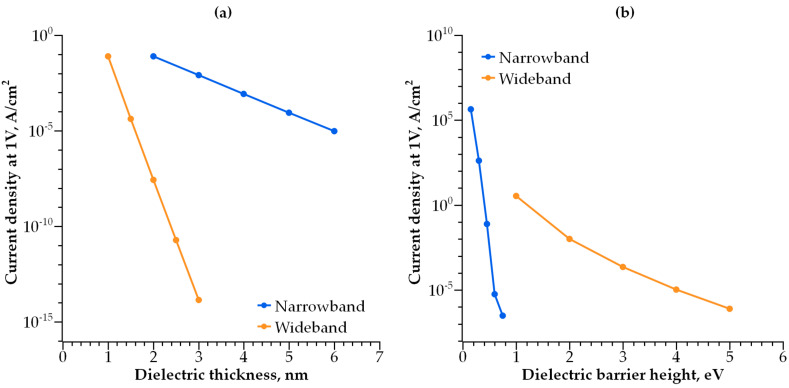
(**a**) Dependence of current density at 1 V on the dielectric thickness; (**b**) current density at 1V dependency on the dielectric barrier height.

**Figure 17 nanomaterials-14-00668-f017:**
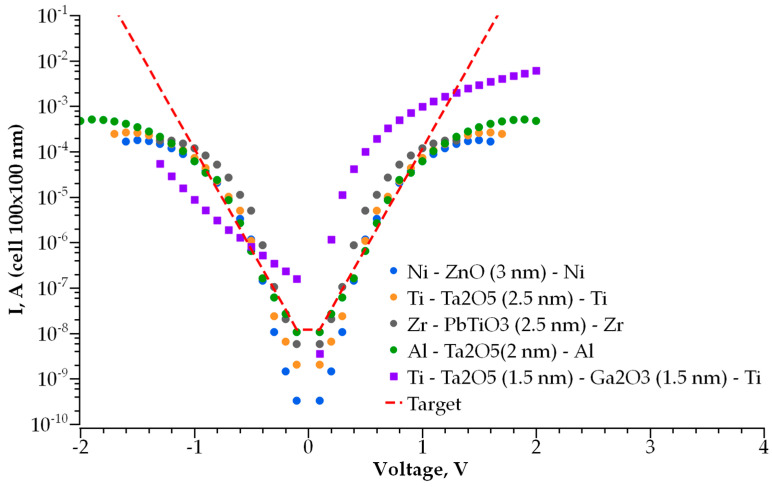
I-V curve of double-layer selector (green dotted line) compared to the best single-layer selectors.

**Figure 18 nanomaterials-14-00668-f018:**
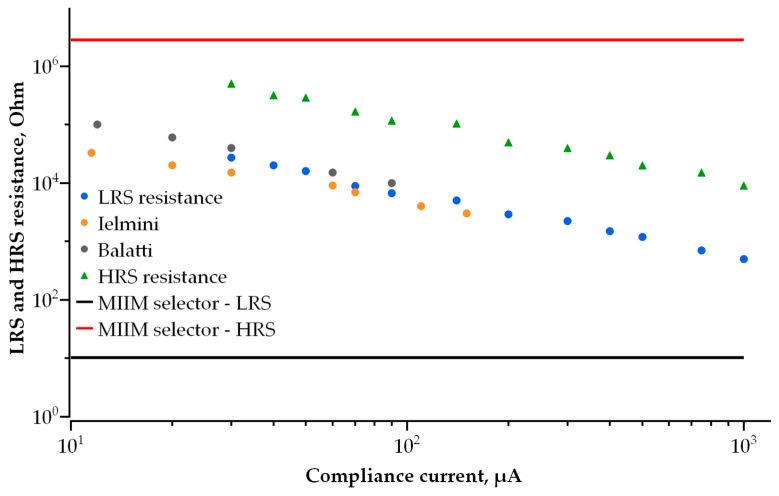
LRS and HRS resistance of double-layer selector device at 1 V and 0.1 V, respectively, as compared to LRS and HRS resistance of RRAM cell.

## Data Availability

The data presented in this study are available on request from the corresponding author.

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
