# Peer review of "Modeling of Self-Aligned Selector Based on Ultra-Thin Metal Oxide for Resistive Random-Access Memory (RRAM) Crossbar Arrays"

_nanomaterials, 2024, doi:10.3390/nano14080668_

Round 1
Reviewer 1 Report
Comments and Suggestions for Authors
structures with single layer and double-layer dielectrics. Comments are below.
1. In Section 2, the authors have mentioned various important properties of selectors. It is suggested to add references related to these specific requirements.
2. In Figure 2, the authors have described the LRS and HRS under different current compliance levels. It is suggested to plot all I-V characteristics related to different current compliance levels, as shown in Figure 2(b).
3. Similarly, authors should provide all I-V characteristics related to different V-RESET stop values.
4. Authors should clearly cite references for equations 1 to 3. Additionally, the authors' contribution to the newly developed simulation must be clearly included.
5. In Figure 7, authors should recheck the experimental and theoretical plots.
6. The presentation of figures must be improved.
Comments on the Quality of English LanguageMust be improved.
Author Response
On behalf of all authors, thank you for your report. You may find our reply in the attached file.

Reviewer 2 Report
Comments and Suggestions for Authors
The study investigates the possibilities of using a tunnel diode based on single- and double-layer dielectrics and determine their optimal physical properties to be used in a HfOx-based RRAM Crossbar array. While the work addresses theoretical and modeling aspects of creating a self-aligned selector with optimal parameters and exhibits some promising aspects, I believe that several key issues must be addressed through major revisions before the manuscript can be deemed suitable for publication. Below, I provide a detailed list of concerns and recommendations for the authors to consider.
Major Concerns and Recommendations:
1. In the Introduction section, the introduction could benefit from a more structured flow. Consider dividing the text into distinct paragraphs, each focusing on a specific aspect (e.g., RRAM advantages, selector requirements, existing selector technologies, the appeal of self-aligned MIM tunnel diodes, and the research context). This would enhance readability and clarify the logical progression of ideas.
2. Derive Equations (4) & (5) in the supplementary material, providing a step-by-step explanation and detailing underlying assumptions for readers' understanding.
3. Verify and correct potential typos in Equations (7) & (8): change to and to .
4. The statement in Chapter 3.1, which posits that "For selectors with a single-layer dielectric, a lower and thicker barrier results in a steeper I-V curve," appears to lack adequate substantiation based on Figure 6. The authors are urged to furnish additional evidence or analytical reasoning to convincingly establish this correlation. This could involve a more detailed analysis of the data presented, supplementary experimental results.
5. There seems to be an inconsistency in the reference to figures within the text. The assertion, "The materials presented in Figure 4 are the most satisfactory dielectrics to be used for self-aligned selectors," might be an error, as it appears more appropriate to refer to Figure 6 instead.
6. A discrepancy has been noted regarding the source attribution for certain content. While equations (1) and (2) are derived from reference [20], the statement " In case of a single-layer dielectric and for low voltages (V < φ), the WKB approximation [18] yields classical equation of direct tunneling:" is attributed to reference 18. This inconsistency raises confusion and requires clarification.
By addressing these concerns and implementing the recommended modifications, the manuscript will significantly improve in terms of clarity, accuracy, and overall scientific rigor, bringing it closer to meeting the standards required for publication in Nanomaterials.
Author Response

(The authors gave the same response as above.)

Reviewer 3 Report
Comments and Suggestions for Authors
Comments on the Quality of English LanguageQuality of English Language is good.
Author Response

(The authors gave the same response as above.)

Round 2
Reviewer 2 Report
Comments and Suggestions for Authors
I am writing to confirm my approval of the revised manuscript titled "Modeling of self-aligned selector based on ultra-thin metal oxide for RRAM Crossbar arrays" for publication in Nanomaterials. The authors have attentively addressed all reviewer comments, resulting in substantial improvements across the board. I am satisfied with the modifications and request that you proceed with the publication process.